# Food Security and Obesity among Mexican Agricultural Migrant Workers

**DOI:** 10.3390/ijerph16214171

**Published:** 2019-10-29

**Authors:** José Castañeda, Graciela Caire-Juvera, Sergio Sandoval, Pedro Alejandro Castañeda, Alma Delia Contreras, Gloria Elena Portillo, María Isabel Ortega-Vélez

**Affiliations:** 1Centro de Investigación en Alimentación y Desarrollo, Hermosillo 83304, Mexico; pepe_jjcp@msn.com; 2Nutrition Division, Centro de Investigación en Alimentación y Desarrollo, Hermosillo 83304, Mexico; gcaire@ciad.mx (G.C.-J.); acontreras@ciad.mx (A.D.C.); gloriela@ciad.mx (G.E.P.); 3Regional Development Division, Centro de Investigación en Alimentación y Desarrollo, Hermosillo 83304, Mexico; ssandoval@ciad.mx; 4Social Sciences Division, University of Sonora, Hermosillo 83000, Mexico; pedroaalejandro@gmail.com

**Keywords:** food insecurity, obesity, migrant agricultural workers, Mexico, public health

## Abstract

Mexican migrant farm workers are one of the poorest and most marginalized social groups within the country. They face the double burden of malnutrition, food insecurity, as well as harsh living and labor conditions. **Objective**: To examine the relationship between household food insecurity (HFI) and obesity in a population of migrant farm workers in highly modernized agribusiness areas of Northwest Mexico. **Methods**: This was a cross-sectional study with a concentric (site) (*n* = 146 households) and systematic selection of participants (adult men and women). Methods included questionnaires regarding socio-demographic characteristics, food security, diet (two non-consecutive 24-h recalls), and physical activity (PA). Anthropometric data included height, weight, and waist circumference. Data analysis covered descriptive statistics, multivariate linear and logistic regression. **Results**: Sample showed 75% prevalence of overweight and obesity, while 87% of households reported some level of HFI. Mild HFI resulted in five times more probability of farm workers’ obesity (OR = 5.18, 95% CI: 1.37–19.58). However, there was a protective effect of HFI for obesity among men (OR 0.089, 95% CI: 0.01–0.58) in a context of intense labor-related PA. **Conclusion**: There is a difference by gender in the relationship of HFI with obesity prevalence related perhaps to the energy expenditure of male agricultural migrant workers.

## 1. Introduction

There is consistent evidence that supports the relationship between obesity and household food insecurity (HFI) in different socioeconomic and cultural contexts. For this purpose, the most referred definition of HFI is that of the USDA as the “limited or uncertain availability of nutritionally adequate and safe foods or limited or uncertain ability to acquire acceptable foods in socially acceptable ways”.

In the U.S., the evidence reported of the obesity–FI relationship is reliable among women, grows among adolescents, produced mixed results among children, and provides scarce evidence among men [1,2]. To explain that consistent association among women, authors have suggested some behavioral and physiological pathways. Among the most mentioned are: Mothers’ strategies to face FI, placing nutritional needs of children and male partners before their own; also, mothers looking for alternative strategies to obtain food for the family when income is not enough can trigger maternal stressors contributing to women’s binge eating and mothers’ low quality diet in FI households. Among the physiological pathways, energy stores as fat for reproduction that are inherent to the female biology, contribute to women’s obesity within FI households [1,3]. In Mexico, Morales-Ruán reported a positive association between FI and obesity among rural, poor, and indigenous women [4]. There is, however, evidence that there are mediators that modify such association; some of them are gender, marital status, stressors, food stamp participation, and diet composition [1,5].

According to the National Nutrition and Health Survey [6,7] in Mexico, 70% of adult women and men are obese or overweight; the survey also reported that 70% of households (almost 50 million people) experienced food insecurity. Furthermore, among the most vulnerable populations in Mexico, migrant agricultural workers migrate mainly from southern Mexico to the more prosperous northwest region of the country. In this region successful export agribusiness hire each year up to 200,000 migrant workers to plant, care for, and harvest a variety of vegetable and fruit crops. The living conditions and health risks of these workers are precarious, as reported in previous publications. These include low wages for seasonal work, and consequently lack of constant and quality access to public health services, which translates to poverty, nutritional deficiencies, and disease [8,9]. Health risks for migrant workers and their families are poverty and work related; they migrate from mild weather regions of the country to regions of extreme weather, and harvest season temperatures go from 45–50 °F (during Winter) to 122 °F (during Summer). In addition, housing is usually precarious, with crowding and poor hygiene. Consequently, dehydration, heat stroke, gastrointestinal and respiratory infectious diseases are common.

Hazardous socioeconomic conditions of a population, such as low seasonal wages, low formal education, dietary changes due to migration, lack of access to health services, and living in an obesogenic environment, could involve a high risk of obesity because of its relationship to food intake of low nutritional quality [10]. Migrant agricultural workers experience dietary changes leading to a rich calorie, fat, sugar, and sodium diets, because of the limited resources to acquire high quality food. There is evidence of the continuous growing prevalence of obesity among these workers, especially among women that have settled in towns close to the farms with respect to those that continue going back and forth between farms and their hometowns (28.1% versus 7.1%; *n* = 402) [9].

It is the aim of this study to analyze the relationship between food security and obesity and its mediators among migrant agricultural workers living in two communities close to produce farms in Northwest Mexico.

## 2. Methods

This was a cross-sectional study, which included 146 adult men and women living in two agricultural communities in Northwest Mexico (Miguel Aleman and Pesqueira, belonging to the municipalities of Hermosillo and San Miguel de Horcasitas, in Sonora, at the northwestern border region of Mexico) (Figure 1). The total sample size was estimated using the proportion of obesity reported by The National Health and Nutrition Survey for Mexican adults [6]. Selection of participants followed a concentric sampling selection of neighborhoods within the two communities [11], a systematic selection of houses within neighborhoods, and a voluntary participation of adult men and women. We counted all houses within each circle and distributed our sample proportionally to the total amount in each. Once the list of addresses was obtained, we used simple random sampling to visit the houses. We chose one point to start the selection of houses at the edge of the first street and selected the first home according to the random number list; if this home was non-inhabited or the respondent did not want to participate, we proceeded to the next household sited to the left of the first house visited. We approached 240 houses to ask for an adult participation; 102 (41%) of them were empty houses or nobody responded to our call, and 12 (5%) refused to participate.

The concentric sampling allowed us to enroll participants accounting for their time of residence in these communities, since those living near to the cultural center had resided there for a longer time, compared to those living far from the cultural center. The selection criteria was 4 years or more of residence in their communities, which is the period proposed by the Ministry of Social Development in Mexico [12], for a migrant to be considered resident. Pregnant or lactating women were not included in the sample. Participants signed a written informed consent, and the Ethics Committee at Centro de Investigación en Alimentación y Desarrollo, A.C. (CE/001/2016, www.ciad.mx) approved the study protocol.

Participant’s interviews included a sociodemographic questionnaire [13], two non-consecutive 24-h recalls [14], a short version of IPAQ questionnaire to measure the level of individual physical activity (PAL) proposed by WHO [15], and a household food security (HFS) scale developed for the Northwest Mexico population [16]. This scale resulted in 15 items categorized as three dimensions, similar to that of the Universal and the Latin-American and Caribbean Scale: Worry or uncertainty for food availability, diet quantity and quality, and socially accepted ways of acquiring food [17,18].

Anthropometric data such as height, weight, and waist circumference were measured using a stadiometer, a digital scale (SECA 50–200 kg ± 0.05–0.1 kg), and a fiberglass measuring tape; the reference to measure waist circumference was the midpoint between the intercostal edge and the iliac crest [19].

We used a food dictionary that included several food composition data banks such as that of the United States Department of Agriculture [20,21], the Mexican National Nutrition Institute [22], and our own data developed at CIAD A.C. of traditional Sonoran dishes [23] to evaluate diet nutrient content. We computed the nutrient content of foods using a method based on an Excel datasheet and a food dictionary [24].

Body mass index (BMI) was calculated from height and weight and classified according to WHO recommendations [25] as wasting (<18.5 kg/m^2^), normal (18.5 to 24.9 kg/m^2^), pre-obesity (25.0 to 29.9 kg/m^2^), and obesity (≥30.0 kg/m^2^). Regarding waist circumference, the cut-off points were 80 to 88 cm for women, and 94 to 102 cm for men [26].

## 3. Results

Table 1 shows participants’ general characteristics. The sample included 83 women (57%) and 63 men (43%). As a population of agricultural workers, most of them were low-income laborers; regarding formal education, 62.3% were illiterate, while 34.3% reported basic formal education, (18.5% elementary school and 15.8% middle school), and 3.4% completed high school. These workers have migrated mainly from southern Mexico localities within the states of Oaxaca, Veracruz, Guerrero, and Chiapas, which have the lowest human development indicators within the country [27]; furthermore, some of the farm workers have labored in several Northwest Mexico farms located in Sinaloa (a border state at southern Sonora), and Chihuahua (at the eastern border of Sonora) (Figure 1). Twenty one percent of workers spoke an indigenous language besides Spanish.

Among female and male participants, 36% were overweight (19% men and 17% women), while 39% were obese (9% men and 30% women). HFI was observed in 87% of families (6% mild HFI, 36% moderate HFI, and 45% severe HFI). Regarding physical activity (PA), 3.2% of men and women showed a mild level (0.8 and 2.4%, respectively), 44% (8.8% men and 35.2% women) were moderately active, while 52.8% were very active (31.2% men and 21.6% women).

### 3.1. Diet

Food intake in at least 70% of the sample included beans, soda pop, fried eggs, sugar, onions, tomatoes, instant coffee, flour tortilla, potatoes, fresh cheese, rice, “salsa” (tomatoes, green chili, onions, and spices), and boiled chicken (33%). Cereals and tubers (corn and flour tortilla) and animal products (boiled chicken and fried eggs) were the main foods contributing to total energy intake (62% for women and 63% for men). Meanwhile fruit, vegetables, and alcohol contributed with the minimal energy intake. Table 2 shows main food or food dishes that contributed to energy intake in each food group. Ultra processed foods contributed to 22% of the total energy intake of farm workers (23.1% for women and 20.9% for men).

### 3.2. HFI and Obesity

Table 3 shows three separated models of association between pre-obesity (BMI 25–30), obesity (BMI > 30), or both, with food insecurity. In the group of obesity, workers living in a household with mild HFI had a five times higher risk of being obese when compared to those living in food secure households (OR = 5.18, 95% CI: 1.37, 19.6). The analysis, however, showed that there is a significant effect of gender in this model; therefore, we explored the odds ratios stratifying for men and women, and the results are in Table 4. In the men’s obesity group, living in households with food insecurity turned out to be a protective factor for obesity, compared to living in households with food security (OR= 0.089, 95% CI: 0.014, 0.058). There was no significant effect of FI on obesity among women, although there is a trend (OR = 0.286, *p* = 0.177) to a protective effect moderated by age, physical activity, and time of residency. Here we should remember that vigorous physical activity was greater in men compared to women, but women showed to be more active than typical Mexican women were.

As seen in Table 5, intense physical activity is a significant variable when exploring the relationship of BMI and HFI. We obtained similar results when waist circumference was considered as the dependent variable (data not shown).

## 4. Discussion

In this study, we observed an association of HFI with obesity; these results are similar to studies in other countries and in Mexican national surveys [4,28,29,30,31,32,33]. We evaluated the possible mediators of the above relationship and found out that intense physical activity could be a modifier of the effect of gender on the association between obesity and food insecurity.

According to our results, migrant farm workers residing in communities close to farms showed a fivefold increase in obesity when they faced mild FI, compared to those in FS households. Several authors have reported this association in the Mexican, American, Colombian, and Brazilian populations, particularly in adult women, but it is not clear in adult men [4,29,30,31,32,33,34]. However, in this study, we found that food insecurity could be a protective factor for obesity in men, but not in women.

There is some evidence of mediators that could modify the association between obesity and food insecurity. Wu et al. [5], reported that this association was mediated by unhealthy dietary behaviors; among them, skipping breakfast was associated with obesity/overweight in females 10 to 18 years (OR = 1.63, 95% CI 1.20–2.22), while consuming snacks and sugared drinks was related to obesity/overweight in males of the same age (OR = 1.51, 95% CI 1.15–1.98). Other authors have also suggested that gender, marital status, stressors, food stamp participation, and diet composition could be mediators on the FI–obesity association [1,5].

In the case of migrant agricultural workers, it seems that poverty is not linearly affecting the probability of developing obesity; social and work context, however, could modify this relationship. On one hand, seasonal work means that there should be uncertainty about food access, related frequently to seasonal work, and lack of income several months a year, which compromises access to food. On the other hand, agricultural work means a high-energy expenditure activity compared to non-agricultural work, even when it is seasonal [35]. According to this, the scarcity hypothesis could explain obesity prevalence in low-income individuals that live with HFI. In conjunction with poverty, HFI could lead to a higher desire for food and a higher perception of food scarcity, or that food scarcity could become a problem in the near future. Furthermore, work seasonality could lead to a high consumption of energy due to a seasonal plentiful access to food. This situation does not appear to be present in higher-income individuals, since uncertainty regarding food availability and access does not exist and neither the stressors that this entails [3].

Authors agree that obesity occurs mainly in poor women in developed and developing countries [36]; poor local contexts moderate the relationship between HFI and obesity. In developing countries, as of Mexico, Morales-Ruán et al. [4] found a positive association between mild HFI and obesity in poor low-educated women from rural and indigenous regions. Meanwhile, Ortiz-Hernández et al. [31,33] reported higher rates of overweight school-age children from households with severe HFI, as well as higher risk of obesity in adolescents from FI households. Authors suggest that food spending in households with FI devoted to purchasing high-energy density low-cost foods that produce satiety, could explain this association. Among those foods were sweet bread, pastries, peanuts, candy, and sugary beverages.

Supporting that evidence, several studies have reported a significant relationship of HFI with poverty, with a high consumption of low-cost, high-energy foods, and with low intake of fruits, vegetables, and fiber [29,32,34]. Some of the reasons for this association are the high cost of foods, lack of nutritional information, or that women with obesity have a different perception regarding food consumption and food quality compared to women that do not present obesity.

On the other hand, most of the published evidence shows an association of moderate and mild HFI and obesity in the presence of poverty. However, the relationship continues controversially. Dhurandhar [3] proposes two main hypotheses to explain causes of this relationship. One of them refers to the intake of a high-energy density and low-cost diet and the other to the limited knowledge, time, and resources that households with FI would face. This author suggested the “scarcity hypothesis”, which proposes that HFI in an environment full of high-energy, low-cost food could cause a positive energy balance, especially in low-income populations.

Regarding physical activity, Jones and Frongillo [37] discussed that it is a predictor of body weight in women and could be associated to HFI. On the other hand, Chaput et al. [38] reported that physical activity seems to be a protective factor against the risk of obesity in FI urban households from Uganda; furthermore, since frequent food scarcity lead to nutritional deficiencies and deteriorate physical status, Saiz et al. [39] found a low probability of adequate physical activity in a U.S. population with HFI. In Mexico, Gutiérrez et al. [6] and Hernández-Avila et al. [7], using national survey data, reported that Mexicans are inactive or sedentary (81.8%); however, for farm workers, physical activity is labor-related as cropping and harvesting demand a large energy expenditure [40]. Among population groups whose moderate or vigorous physical activity is part of their labor (daily or seasonal), there could be a moderator effect on the association between HFI and obesity. Food quality and quantity plus intense workloads could lessen obesity development, even if HFI is present. The limited power of our sample did not allow a deep data analysis of variables that determine gender differences in the relationship of FI and obesity prevalence. However, the differences by gender seem to be because, in this group of study, men are more physically active than women.

There are some other explanations proposed to the HFI–obesity paradox. Studies have reported inconsistent results when examining the relationship of HFI and prevalence of overweight and obesity in men; authors’ explanations include the nature of the HFS scale, biological sex differences, or gender behaviors. Smith et al. [41] found an association of HFI with obesity in Mexican-American women but not men; authors suggested that maternal behaviors regarding children’s feeding could mediate the association, since the study was not exclusive to parents. On the other hand, Hernández et al. [2] discussed two possible mechanisms related to gender disparities observed in the food insecurity and overweight/obesity relationship. One is how the food security scale captures the responses of women and men; authors propose that there seems to be a different perception of men over the severity of HFI, since they could not be as aware of the uncertainty of food availability as women are. According to these authors, this perceived uncertainty could mean different levels of anxiety regarding food scarcity and may contribute to differences in physiological responses to stress that are associated with obesity. In addition, women could be more aware of strategies to face HFI such as sending children to eat at relatives’ homes, pawning utensils to buy food, or make changes in the preparation and distribution of food within the family. All that may affect food quantity and quality [2].

Biological explanations for the HFI–obesity relationship that have been suggested include that fattening is a physiologically regulated response to food scarcity, especially in low-social status individuals; consequently, there is an increased risk of obesity [3,36]. In the case of gender differences, women are more vulnerable to unhealthy eating habits and having lower sedentary metabolic rates, which can lead them to display excessive weight gain, compared to men.

## 5. Conclusions

This study suggests that physical activity could mediate gender differences on the association between HFI and obesity, especially in individuals with intense labor-related physical activity, living in a context of a developing country; this means living in poverty and having a high-energy, low-cost diet. However, we need a deeper understanding on gender differences in eating behaviors, strategies to face HFI, and biological mediators of weight gain in this population.

## 6. Strengths and Limitations

We recognize some limitations of this study. First, because we used a cross-sectional design, cause–effect relationship between FI and obesity cannot be determined; second, our sampling strategy does not significantly represent men and women; instead, sample characteristics resulted from a systematic selection of households. Moreover, our sample size, although it was calculated according to the prevalence of obesity in Mexican adults, could be small to address some of the statistical tests we used. Among the strengths, we consider that the specific characteristic of the sample, labor-related intense physical activity, allowed us to examine a population lifestyle activity that goes out of the typical trend, specifically among Mexican low-income population.

## Figures and Tables

**Figure 1 ijerph-16-04171-f001:**
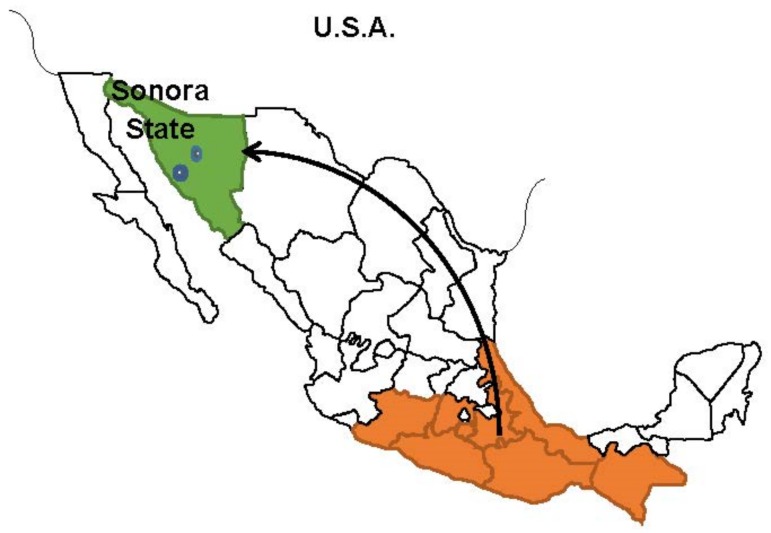
Regions of attraction and origin of migrant farm workers in Mexico.

**Table 1 ijerph-16-04171-t001:** General characteristics of migrant agricultural workers in Miguel Aleman and Pesqueira, Sonora at northern Mexico.

Variable	Women [*n* = 83]		Men [*n* = 63]	
	X ± DS	Range	X ± DS	Range
Age (years)	41.9 ± 12.5	18–68	44.7 ± 11.9	24–68
Time of residency (years)	20.9 ± 11.8	4–61	19.7 ± 11.9	4–53
Weight (kg)	71.8 ± 17.5	41.5–127.8	70.9 ± 10.9	52.6–99.1
Height (m)	1.5 ± 0.1	1.3–1.7	1.6 ± 0.1	1.5–1.8
BMI (kg/m^2^)	31.2 ± 7.2	18.8–55.7	26.8 ± 4.3	18.5–38.4
Waist circumference (cm)	97.4 ± 15.7	67.6–150.0	92.4 ± 9.9	69.1–110.9

**Table 2 ijerph-16-04171-t002:** Main energy-supplying foods per food group.

Food Group	Food Group Contribution to Total Energy (%)	Main Foods
Cereals and tubers	40.6	Corn and flour tortilla
Animal products	22.1	Boiled chicken and fried eggs
Sugar, sugared drinks	9.2	Soda pop, sugar
Oils and fats	5.9	Vegetable oil, “chorizo”/Mexican sausage
Legumes	1.7	Beans
Ultra processed foods/sugar and fat added	5.9	Fried corn tortilla, sweet bread, ham pizza
Fruit	2.6	Tamarind, mangoes, watermelon, oranges
Vegetables	1.7	Tomatoes, onions
Milk and yogurt	1.0	Whole milk, fruit yogurt
Alcoholic beverages	0.6	Beer
Non-energy foods	1.3	Broth

**Table 3 ijerph-16-04171-t003:** Odds Ratios and 95% CI for the association between each obesity group and Household Food Insecurity (HFI) among migrant farm workers in Northwest Mexico.

Variable	Food Security *n* = 19	Mild HFI *n* = 9			Moderate HFI *n* = 54			Severe HFI *n* = 64		
		OR	*p*	95% CI	OR	*p*	95% CI	OR	*p*	95% CI
Pre-obesity (*n* = 53)	Reference	0.986	0.989	0.142–6.86	0.697	0.698	0.112–4.33	0.526	0.271	0.168–1.65
Obesity (*n* = 56)	Reference	5.180	0.015	1.37–19.58	0.275	0.283	0.026–2.90	1.002	0.996	0.384–2.61
Pre-obesity + obesity (*n* = 108)	Reference	3.081	0.183	0.588–16.14	0.453	0.383	0.076–2.68	0.623	0.356	0.228–1.70

FI: Food insecurity. OR: Odds Ratios. 95% CI: 95% Confidence interval. Models adjusted by gender, age, physical activity, and time of residency (*p* ≤ 0.05).

**Table 4 ijerph-16-04171-t004:** Odds Ratios and 95% CI between each obesity group and food insecurity level among male and female migrant farm workers in Northwest Mexico.

Variable	FS (*n* = 8)	FI (*n* = 75)		
		OR	*p*	95% CI
**Female**				
Pre-obesity (*n* = 25)	Reference	1.292	0.863	0.071–23.65
Obesity (*n* = 43)	Reference	0.286	0.177	0.046–1.762
Pre-obesity + obesity (*n* = 68)	Reference	0.598	0.543	0.056–5.234
**Male**				
Pre-obesity (*n* = 28)	Reference	0.814	0.880	0.056–11.81
Obesity (*n* = 13)	Reference	0.089	0.011	0.014–0.0575
Pre-obesity + obesity (*n* = 41)	Reference	0.180	0.131	0.019–1.671

FS: Food security, FI: Food insecurity OR: Odds Ratio. 95% CI: Confidence intervals to 95%. Model adjusted by sex, age, physical activity, and time of residency (*p* ≤ 0.05).

**Table 5 ijerph-16-04171-t005:** Effect of HFI, physical activity, age, and time of residency on body mass index (BMI) of male and female farm workers from Northwest Mexico.

Variable	Coefficient	*p*	95% CI
**Female (*n* = 74)**			
Constant	18.867	0.003	6.649–31.085
Age	0.132	0.115	−0.033–0.296
Mild HFI	−3.835	0.360	−12.141–4.470
Moderate HFI	−0.708	0.805	−6.403–4.986
Severe HFI	−0.370	0.895	−5.937–5.198
Moderate PAL	6.840	0.127	−1.985–15.664
Intense PAL	4.388	0.331	−4.559–13.335
Time of residency	0.088	0.289	−0.076–0.251
**Male (*n* = 51)**			
Constant	37.661	0.000	27.877–47.445
Age	−0.007	0.904	−0.128–0.113
Mild HFI	−3.995	0.202	−10.210–2.221
Moderate HFI	−4.597	0.010	−8.019–(−1.175)
Severe HFI	−3.260	0.037	−6.313–(−0.207)
Moderate PAL	−5.702	0.188	−14.292–2.888
Intense PAL	−8.145	0.050	−16.291–0.002
Time of residency	−0.010	0.854	−0.120–0.099

FS: Food security, FI: Food insecurity. 95% CI: Confidence intervals to 95% (*p* ≤ 0.05).

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
