# Peer review of "Food Security and Obesity among Mexican Agricultural Migrant Workers"

_ijerph, 2019, doi:10.3390/ijerph16214171_

Round 1

Reviewer 1 Report

This is an interesting paper on the relation between obesity and food insecurity in Mexican population. I make some suggestions that may help improving the manuscript.

Introduction:

- The introducion is too concise. My advice to make is stronger would be to include information on the following:

a) Refer to the patways that have been suggested to link the relationship between obesity and FI (beyond what is explained in lines 40-42)

b) p 2. line 49 says: "the living conditions and health risks of these workers are precarious, as reported in previous publications". Please, provide more detail on these living conditions and health risks. 

c) Describe the "hazardouns socioeconomic conditions" that trese migrants face in terms of income, education and other relevant aspects. 

Methods:

- I am not familiar with the term "concentric sampling" nor have I been able to quickly find an explanation of the concept online. Can you explain and reference it?

Results:

- What is the percentage of men and women in the sample?

- You mention the overall prevalence of HFI. I think it would be useful for the purpose of the study to show (in a table?) the results of the 15items and 3 dimensions, as it will allow to better discuss the pathways mediating the relation between OB and FHI. 

- Line 145, please also discuss the results for women.

Overall, English language needs to be reviewed.

Author Response

Reviewer 1.

Introduction

Reviewer: The introduction is too concise. My advice to make is stronger would be to include information on the following:

Refer to the pathways that have been suggested to link the relationship between obesity and FI (beyond what is explained in lines 40-42).

Reply: this paragraph has been added following lines 40-42:  To explain that consistent association among women, authors have suggested some behavioral and physiological pathways. Among the most mentioned are: mother's strategies to face FI, placing nutritional needs of  children and male partners before their own; also, mother's looking for alternative strategies to obtain food for the family when income is not enough,  can trigger maternal stress contributing to women’s binge eating and mother's low quality diet in FI households. Among the physiological pathways, energy stores as fat for reproduction that are inherent to the female biology, contribute to women's obesity within FI households. (Nettle et al., 2017; Franklin et al., 2012; Dhurandhar, 2016).

p 2. Line 49 says: “the living conditions and health risks of these workers are precarious, as reported in previous publications”. Please, provide more detail on these living conditions and health risks.

Reply: To address reviewer suggestion we added the following paragraph: These include low wages for a seasonal work, and consequently lack of constant and quality access to public health services, which translates in poverty, nutritional deficiencies and disease [7, 8]. Health risks for migrant workers and their families are poverty and work related;  they migrate from mild weather regions of the country, to regions of extreme weather and harvest time temperatures go from 45-50 °F (during Winter) to 122 °F during Summer). In addition, housing is usually precarious, with crowding and poor hygiene. Consequently, dehydration, heat stroke, gastrointestinal and respiratory infectious diseases are common.

Describe the “hazardous socioeconomic conditions” that these migrants face in terms of income, education and other relevant aspects.

Reply: To address reviewer suggestion we added the following paragraph: Hazardous socioeconomic conditions of a population, such as low seasonal wages, low formal education, dietary changes due to migration, lack of access to health services, and living in an obesogenic environment, could involve a high risk of obesity because of its relationship to food intake of low nutritional quality.

Methods:

Reviewer:        -I am not familiar with the term “concentric sampling” nor have I been able to quickly find an explanation of the concept online. Can you explain and reference it?

Reply: Concentric sampling was a way of selecting subjects that one of the research team member proposed. His discipline is that of the Anthropology, and the concept comes from Ernest Burgess from the School of Chicago (Figure in following address)

http://ecologiaurbanayrural.blogspot.com/2011/07/modelos-urbanos-de-estructura-interna.html

This methodology allowed us to enroll participants from different migration periods, but that have settled in towns close to farms. People that came to the town 40 years ago, used to live in the social center of the town, meanwhile those that came recently (we used an official time of 4 years to be considered resident), used to live in the towns periphery. The two towns in which we collected data have developed around the main City of Hermosillo in the last 50 years, mainly by migrant people  that came to work in farms close to the towns settlements  (from nearby states in the beginning and afterwards from population from southern Mexico in the last 20 years).

We include a paper from 2010 that explain the methodological framework and a view of the School of Chicago theoretical contributions in google drive shared with  [email protected]

Results:

                -What is the percentage of men and women in the sample?

Reply: It has been added in this paragraph: Table 1 shows participants’ general characteristics. Sample included 83 women (57%) and 63 men (43%).

                -You mention the overall prevalence of HFI. I think it would be useful for the purpose of the study to show (I a table?) the results of the 15 items and 3 dimensions, as it will allow to better discuss the pathways mediating the relation between OB and HFI.

Reply: Thank you for this suggestion. We are working on it.

                -Line 145, please also discuss the results for women.

Reply: This paragraph has been added to the results section mentioned: There was no significant effect of FI on obesity among women, although there is a trend (OR = 0.286, p = 0.177) to a protective effect moderated by age, physical activity, and time of residency). Here we should remember that vigorous physical activity was greater in men compared to women, but women showed to be more active than typical Mexican women were.  

Reviewer: Overall, English language needs to be reviewed:

Reply: We will do so.

Reviewer 2 Report

Thank you very much for this paper. This is a very important issue and this topic has not been deeply explored in such a specific population subgroup. So very relevant issue. There are, however, several comments that I have regarding your work:

Introduction

You explain well the rationale of the study. However, you should explain what food insecurity is. Depending on the reader, their prior knowledge about the topic might vary, so a definition is needed. There are several typos and grammar errors. A review of the English should be considered

Methods

3. About the sampling method: You mention that you used a concentric sampling, but you do not mention how many "circles" you used to divide the towns. You also don't mention how many households came from each "circle" (e.g. how many from the city centre, how many from the outskirts etc.)

4. You mentioned that you used a systematic selection of houses. You should describe which system you used to select the houses and how many houses you selected.

5. You should also mention the response rate: number of people who did the questionnaire/ number of people invited to participate.

6. Your aim is to study migrant workers, but in your selection criteria you include everyone living in these cities for over 4 years. This means that some people in your sample, according to table 1, had been living there for 50 or 60 years. How do you know they were all migrants? None of them were born in these cities? If not everyone was a migrant, you need to rephrase the whole article and stop talking about migrant workers, but individuals living in a "migrant community".

7. You describe well how you have constructed the questionnaire; and you have used validated tools, which I think it is good. However, you should make the questionnaire available, probably by including it in the supplementary material.

8. Equally, the method described in page 3 line 103 used to compute the nutrient content of foods should be available in the supplementary material.

Results

9. Table 2.  I would suggest to include the % that each food group contributed to the energy intake of participants.

10. Section 3.2. (page 5, line 139). You divide the models in "pre-obesity" and "obesity". You should explain what pre-obesity means. Is it a BMI 25-30? It so, it should be stated

Discussion

11. You explain the protective effect of HFI in men by saying that physical activity mediates the relationship. If so, you would expect higher levels of physical activity in those with high FI, you would also expect higher levels of FI in those with lower income. Given that you have the data needed this is something you should consider presenting. Otherwise is just an assumption.

12. Page 8 line 183, you say that "migrant farmworkers residing in communities close to farms showed a five times increase in BMI when they faced mild FI". Looking at your data this is not true, it should be that they are five times more likely to be obese.

13. Page 8 like 185. You mention "Following this line of thought". But you are not following the line of thought, you are mentioning the opposite of what you were mentioning before.

14. The paragraph starting on page 8 line 188 is very unclear and should be reworded to explain things better. At the beginning you mention that FI is mediated by unhealthy behaviours, but then you say that FI goes beyond dietary behaviours. That's fine as a concept, but it is not well explained at all.

Conclusions

15. I think your conclusions are not sustained by your results. You have not proven that physical activity mediates gender differences between HFI and obesity. For that, you should prove that physical activity related with HFI, which is not present in the results right now. You data seems to show that, specially in men, HFI is not related with obesity, actually the opposite. But your data is too small to reach definite conclusions.

Strengths and limitations

16. You should mention that the small size of your sample (lack of power) is also a limitation. Another limitation is that you did not use random methods of sampling.

Author Response

Thank you very much for this paper. This is a very important issue and this topic has not been deeply explored in such a specific population subgroup. So very relevant issue. There are, however, several comments that I have regarding your work:

Introduction

You explain well the rationale of the study. However, you should explain what food insecurity is. Depending on the reader, their prior knowledge about the topic might vary, so a definition is needed. There are several typos and grammar errors. A review of the English should be considered

Reply: The definition has been added at the beginning of the introduction as follows: . For this purpose, the most referred definition of HFI is that of the USDA as the “limited or uncertain availability of nutritionally adequate and safe foods or limited or uncertain ability to acquire acceptable foods in socially acceptable ways”.  

We will send the paper to the journal language editing services.

Methods

About the sampling method: You mention that you used a concentric sampling, but you do not mention how many "circles" you used to divide the towns. You also don't mention how many households came from each "circle" (e.g. how many from the city centre, how many from the outskirts etc.)

Reply: In each community (Miguel Alemán, total population: 22,505, 51.4% male and 48.6% female; and Pesqueira, Total population: 3648, 52% male and 48% female), we defined three circles that represented outskirts (more recent migration), a middle circle and the institutional, commercial and social center of the town, located actually close to the main city entrance and exit road. Figures were included as a supplement.

You mentioned that you used a systematic selection of houses. You should describe which system you used to select the houses and how many houses you selected.

Reply:  We added the following to the paragraph: We counted all houses within each circle and distributed our sample proportionally to the total amount in each. Once the list of addresses was obtained, we used simple random sampling to visit the houses. We chose one point to start the selection of houses at the edge of the first street, and selected the first home according to the random number list; if this home was non-inhabited or the respondent did not want to participate, we proceeded to the next household sited to the right side of the first house visited.

You should also mention the response rate: number of people who did the questionnaire/ number of people invited to participate.

Reply: We approached 240 houses to ask for an adult participation; 102 (41%) of them were empty houses or nobody responded to our call, and 12(5%) refused to participate.

Your aim is to study migrant workers, but in your selection criteria, you include everyone living in these cities for over 4 years. This means that some people in your sample, according to table 1, had been living there for 50 or 60 years. How do you know they were all migrants? None of them were born in these cities? If not everyone was a migrant, you need to rephrase the whole article and stop talking about migrant workers, but individuals living in a "migrant community".

Reply: The two settlements originated and grew historically through a migration process related to the growth of commercial agriculture in the zone. In fact, the amount of inhabitants almost doubles during the harvesting period. In previous studies, we found that migrants that have settled in town (considering more than 4 years of residency according to the Social Development Secretariat, a governmental institution) develops more than twice the prevalence of obesity than recent migrants. That is the reason why we focused on settled migrants, which still work as a farmworkers. In addition, in our logistic regression analysis, we included the time of residency as an adjusting variable, precisely because of the wide range of years of migration

You describe well how you have constructed the questionnaire; and you have used validated tools, which I think it is good. However, you should make the questionnaire available, probably by including it in the supplementary material.

Reply: We will include a document with a table with the items that are part of the FS questionnaire as a supplementary document, and as the Reviewer 1 has suggested, we will examine the proportion of responses according to prevalence of obesity.

Equally, the method described in page 3 line 103 used to compute the nutrient content of foods should be available in the supplementary material.

Reply: We will include part of the manual that we developed at CIAD.

Results

Table 2.  I would suggest to include the % that each food group contributed to the energy intake of participants.

Reply: We did it..

Section 3.2. (page 5, line 139). You divide the models in "pre-obesity" and "obesity". You should explain what pre-obesity means. Is it a BMI 25-30? It so, it should be stated.

Reply: Yes, pre-obesity is a BMI 25-30. We included the description as follows: pre-obesity (BMI 25-30), obesity (BMI >30),

Discussion

You explain the protective effect of HFI in men by saying that physical activity mediates the relationship. If so, you would expect higher levels of physical activity in those with high FI, you would also expect higher levels of FI in those with lower income. Given that you have the data needed this is something you should consider presenting. Otherwise is just an assumption.

Reply: The following data can illustrate the relation of physical activity to food security and insecurity.

Physical activity

(n = 125)

HFS (% of households)

HFI (% of households)

Female

  Mild PAL

0

2.4

  Moderate PAL

3.2

32.0

  Intense PAL

3.2

18.4

Male

  Mild PAL

0

0.8

  Moderate PAL

3.2

5.6

  Intense PAL

4.8

26.4

Some of the data on physical activity were missing, then n =125

And, food security and insecurity with socioeconomic level

Socioeconomic level

(n = 146)

HFS (% of households)

HFI (%(% of households)

Female

Low

5.5

50.0

Medium low

0.0

1.4

Male

Low

6.2

35.6

Medium low

1.4

0.0

In addition, Table 5 shows that intense physical activity level (PAL) has a significant effect on the association of FI and BMI.

Page 8 line 183, you say that "migrant farmworkers residing in communities close to farms showed five times increase in BMI when they faced mild FI". Looking at your data this is not true, it should be that they are five times more likely to be obese.

Reply: Reviewer is right. That could have happened because some models that we proved before were multiple regression analysis with BMI as the dependent variable…we decided later to use logistic regression. We corrected this in the indicated paragraph.

Page 8 like 185. You mention "Following this line of thought". But you are not following the line of thought, you are mentioning the opposite of what you were mentioning before.

Reply: Thank you for noticing this. We changed that paragraph to: However, in this study, we found that food insecurity could be a protective factor for obesity in men, but not in women.

The paragraph starting on page 8 line 188 is very unclear and should be reworded to explain things better. At the beginning you mention that FI is mediated by unhealthy behaviours, but then you say that FI goes beyond dietary behaviors. That's fine as a concept, but it is not well explained at all.

Reply: Reply: We re-wrote this paragraph in order to add clarity, however we will look for a formal English editing of the entire document.  See below.

There are some evidence of mediators that could modify the association between obesity and food insecurity. Wu et al. [4], reported that this association was mediated by unhealthy dietary behaviors; among them, skipping breakfast was associated with obesity/overweight in females 10 to 18 years (OR=1.63, 95% CI 1.20–2.22), while consuming snacks and sugared drinks was related to obesity/overweight in males of the same age (OR=1.51, 95% CI 1.15–1.98). Other authors have also suggested that gender, marital status, stressors, food stamp participation, and diet composition could be mediators on the FI-obesity association [1, 4].

Conclusions

I think your conclusions are not sustained by your results. You have not proven that physical activity mediates gender differences between HFI and obesity. For that, you should prove that physical activity related with HFI, which is not present in the results right now. You data seems to show that, especially in men, HFI is not related with obesity, actually the opposite. But your data is too small to reach definite conclusions.

Reply: We think that the model to explain BMI in men and women of our sample in Table 5, suggests that Moderate HFI and Severe HFI are associated significantly with lower BMI. In this model physical activity level is kept in the model since p≤0.05, explaining that the relationship between BMI and HFI is modulated by intense physical activity in men. Definitely, this could be better explored with a bigger sample size.

Strengths and limitations

You should mention that the small size of your sample (lack of power) is also a limitation. Another limitation is that you did not use random methods of sampling.

Reply: We addressed the first suggestion in the limitations section.

About the second suggestion, we added a paragraph in the methods section because, in fact, we performed a random sampling of houses:

Once the list of addresses was obtained, we used simple random sampling to visit the houses. We chose one point to start the selection of houses at the edge of the first street, and selected the first home according to the random number list; if this home was non-inhabited or the respondent did not want to participate, we proceeded to the next household sited to the left of the first house visited.

Thank you very much for your comments and suggestions.

Reviewer 3 Report

Quite an interesting paper on food insecurity and obesity, but it requires some additional support in the introduction and discussion.

Since there is data available from the National Survey, a baseline reference to obesity status in previous years in the regions of interest could be made in relation to income and occupation to show any relevant trends.

Referring to the gender differences, there is no discussion about male and female obesity types which relate to specific metabolic pathways that are to a large degree independent from nutrition and food insecurity.

The waist to hip ratio is a good obesity indicator that takes into account body fat distribution and may show some additional light into the observed associations.

Also to consider are indicators such as cultural beliefs, especially regarding female obesity as in some population groups this is linked to prosperity, fertility and attraction.

Finally, some English language editing is required, e.g. line 45 experience NOT experiment, l 186 thought NOT though, l 188 there is NOT are, l 207 neither DO, l 221 do not PRESENT obesity, l 223 continues to  be controversial, l 240 did not allow, l 242 delete ARE etc   

Author Response

Quite an interesting paper on food insecurity and obesity, but it requires some additional support in the introduction and discussion.

Reviewer: Since there is data available from the National Survey, a baseline reference to obesity status in previous years in the regions of interest could be made in relation to income and occupation to show any relevant trends.

Reply: The National Survey on Nutrition and Health (ENSANUT) gives us clustered data by main cities (Hermosillo). The communities that we sampled are settlements that officially belong to a main city (Miguel Aleman to Hermosillo City and Pesqueira to San Miguel de Horcasitas). We cannot have data specifically of our study population from the National Survey. However, we have collected data on nutritional status that represent these populations in previous studies. Results have shown that migrant settled farmworkers show higher prevalence of obesity than mobile farmworkers. That is the reason we decided to work with migrant farmworkers that have settled. In page 2, paragraph 3, lines 51 to 57 we included some of those data.

Following you can find some previous publications on the matter.

Ortega-Vélez M.I., P.A. Castañeda-Pacheco. 2018.  Ambiente alimentario y seguridad nutricional entre jornaleros migrantes en Sonora. Boletín Científico Sapiens Research. Vol. 8(2). Pp. 18-28. ISSN-e: 2215-9312.

Cecilia Rosales, Maria Isabel Ortega, Jill Guernsey De Zapien, Alma Delia Contreras Paniagua, Antonio Zapien, Maia Ingram and Patricia Aranda.  2012. The US/Mexico Border: A Binational Approach to Framing Challenges and Constructing Solutions for Improving Farmworkers’ Lives. Int. J. Environ. Res. Public Health, 9, 2159-2174.

Ortega MI y Castañeda Pedro Alejandro. Los jornaleros agrícolas en Sonora: condiciones de nutrición y salud. 2007. En Jornaleros Agrícolas Migrantes en el noroeste de México. Ma. Isabel Ortega Vélez, Pedro Alejandro Castañeda Pacheco y Juan Luis Sariego Rodríguez, Coordinadores. Ed.  Plaza y Valdez. Pp. 145-158.

Reviewer: Referring to the gender differences, there is no discussion about male and female obesity types which relate to specific metabolic pathways that are to a large degree independent from nutrition and food insecurity.

Reply: We address this issue at the end of our discussion; page 10, second paragraph, lines 261-263. However, we recognize that this should be deeply explored in future studies.

Reviewer: The waist to hip ratio is a good obesity indicator that takes into account body fat distribution and may show some additional light into the observed associations.

Reply: in this study, we measured waist circumference in addition to BMI, since we have evidence that is a more suitable indicator for adiposity in our population, than the waist/ hip ratio. However, BMI was a better predictor in our logistic regression analysis.

Reviewer: Also to consider are indicators such as cultural beliefs, especially regarding female obesity as in some population groups this is linked to prosperity, fertility and attraction.

Reply:  While it is true that Mexican culture has historically addressed those beliefs, the emergence of obesity that was detected first in the national survey of 2006, has changed the perception of a good part of the population. This happened because health systems have pointed out through national campaigns and in direct care in hospitals and clinics, that obesity conveys a high health risk 

Reviewer: Finally, some English language editing is required, e.g. line 45 experience NOT experiment, l 186 thought NOT though, l 188 there is NOT are, l 207 neither DO, l 221 do not PRESENT obesity, l 223 continues to  be controversial, l 240 did not allow, l 242 delete ARE etc   

Reply: We will send the paper to the journal language editing service.

Thank you very much for your comments and suggestions.

Round 2

Reviewer 2 Report

My comments have been addressed. Many thanks